# Preparation and Characteristics of an Environmentally Friendly Hyperbranched Flame-Retardant Polyurethane Hybrid Containing Nitrogen, Phosphorus, and Silicon

**DOI:** 10.3390/polym11040720

**Published:** 2019-04-19

**Authors:** Chin-Hsing Chen, Chin-Lung Chiang

**Affiliations:** 1Department of Chemical and Materials Engineering, Chinese Culture University, Yang-Ming-Shan, Taipei City 11114, Taiwan; cjx@faculty.pccu.edu.tw; 2Green Flame Retardant Material Research Laboratory, Department of Safety, Health and Environmental Engineering, Hung-Kuang University, Taichung 433, Taiwan

**Keywords:** polyurethane, sol–gel method, hyperbranched hybrid, thermal stability, flame retardant

## Abstract

The NCO functional group of 3-isocyanatoproplytriethoxysilane (IPTS) and the OH functional group of 10-(2,5-dihydroxyphenyl)-10H-9-oxa-10-phospha-phenantbrene-10-oxide (DOPO-BQ) were used to conduct an addition reaction. Following completion of the reaction, triglycidyl isocyanurate (TGIC) was introduced to conduct a ring-opening reaction. Subsequently, a sol–gel method was used to initiate a hydrolysis–condensation reaction on TGIC–IPTS–DOPO-BQ to form a hyperbranched nitrogen–phosphorous–silicon (HBNPSi) flame retardant. This flame retardant was incorporated into a polyurethane (PU) matrix to prepare a hybrid material. Fourier-transform infrared spectroscopy (FTIR), thermogravimetric analysis (TGA), limiting oxygen index (LOI), UV-VIS spectrophotometry, and Raman analysis were conducted to characterize the structure and analyze the transparency, thermal stability, flame retardancy, and residual char to understand the flame retardant mechanism of the prepared hybrid material. After the flame retardant was added, the maximum degradation rate decreased from −36 to −17 wt.%/min, the integral procedural decomposition temperature (IPDT) increased from 348 to 488 °C, and the char yield increased from 0.7 to 8.1 wt.%. The aforementioned results verified that the thermal stability of PU can be improved after adding HBNPSi. The LOI analysis indicated that the pristine PU was flammable because the LOI of pristine PU was only 19. When the content of added HBNPSi was 40%, the LOI value was 26; thus the PU hybrid became nonflammable.

## 1. Introduction 

A reaction of isocyanate and polyhydric alcohol can form polyurethane (PU). It is a kind of polymer with high crosslinking and specific characteristics that are suitable for various types of materials [1,2,3], such as household mattresses, furniture, sealants, adhesives, leather, and various types of biomedical products [4]. However, PU is highly flammable because it contains considerable amounts of aliphatic segments. The mechanical properties of PU deteriorate rapidly if the temperature exceeds 80–90 °C and serious thermal degradation occurs if the temperature exceeds 200 °C, thus its application in many fields is greatly limited. Therefore, application of effective and environmentally friendly flame retardants in PU is crucial [5,6,7,8,9,10]. 

Halogen-free flame retardants are currently receiving increasing attention; compounds that include phosphorus, nitrogen, and silicon are gradually replacing halogen flame retardants. A phosphorus-based flame retardant has two noteworthy characteristics. First, it can be volatilized to form phosphorus free radicals in a gas phase, which can capture free radicals produced from the materials in the fire. Second, phosphate flame retardants can catalyze char formation. Both of these characteristics of phosphorus flame retardants can stop combustion [11]. Additionally, with Si–O bonds as the main chain, siloxane has relatively high bond energy and marked flexibility of Si–O chain segments, which contribute to excellent thermal stability under high temperatures [12].

The PU used in this study has several desirable characteristics, but all PUs have several shortcomings such as flammability, low melting point, and low heat tolerance. Therefore, the sol–gel method was used to prepare a flame retardant that contained nitrogen, phosphorus, and silicon, and combined it with PU to form a hybrid material. The used method is based on the so-called aqueous sol–gel technology, which is generally divided into two steps: the first one named hydrolysis, which produces hydroxyl groups, and the second one named condensation, that involves the polycondensation of hydroxyl groups and residual alkoxyl groups to form three-dimensional hyperbranched particles [13,14,15,16]. Additionally, Fourier-transform infrared spectroscopy (FTIR) was used to conduct structure characterization. The thermostability of this hybrid material was measured through thermogravimetric analysis (TGA), integral procedural decomposition temperature (IPDT), and TGΔ. The flame retardancy of the hybrid material was determined through the limiting oxygen index (LOI). A Raman spectrum was used to study the mechanism of flame retardancy from char analysis. 

## 2. Experimental

### 2.1. Materials

Isophorone diisocyanate (IPDI) and dibutyltin dilaurate (DBTDL) were purchased from Alfa Aesar Co., Shore Road, Heysham, LA3 2XY, UK. ARCOL Polyol 1021(polyol) was purchased from Bayer Material Science Taiwan Limited. Taipei, Taiwan. Furthermore, 1,4-butanediol (1,4-BD) was purchased from Acros Chemical Co., Springfield Township, NJ, USA and 3-isocyanatopropyltriethoxysilane (IPTS) was purchased from Gelest, Inc., 11 Steel Rd. East, Morrisville, PA, USA. Triglycidyl isocyanurate(TGIC) was purchased from Tokyo Chemical Industry(TCI) Co., Ltd, Tokyo, Japan; while, 10-(2,5-dihydroxyphenyl)-10H-9-oxa-10-phospha-phenantbrene-10-oxide (DOPO-BQ) was purchased from Sigma-Aldrich Co. Ltd., Taipei, Taiwan. Anhydrous stabilized tetrahydrofuran (THF) was obtained from Lancaster Co., Morecambe, Lancashire, UK. Hydrogen chloride (HCl) was purchased from Echo Chemical Co., Ltd., Toufen Chen, Miaoli, Taiwan.

### 2.2. Preparation of DOPO-BQ–IPTS–TGIC

First, DOPO-BQ (7.58 g) and IPTS (5.78 g) were added into a 100 mL serum bottle, and then 80 mL of THF solvent and 0.2 g of metal catalyst DBTDL were added. The aforementioned substances were stirred at 60 °C and the substances were allowed to react for 2 h. Subsequently, we added 0.72 mL of DI water into THF, instilled HCl, and adjusted the pH value to 4; the product of the sol–gel reaction was HBPSi. Additionally, TGIC (6.95 g) was dissolved in THF and a sol–gel reaction was conducted for 2 h on HBPSi. After the temperature reached 60 °C, TGIC was slowly added into HBPSi, and allowed to react at 60 °C for 2 h. This formed a hyperbranched nitrogen–phosphorous–silicon (HBNPSi) flame retardant which was dissolved in THF, and this was followed by the addition reaction with a prepolymer to form nanocomposites. Its reaction is presented in Scheme 1. 

### 2.3. Preparation of PU/HBNPSi Hybrid

IPDI (9.88 g) and polyol (20 g) were added into a four-necked bottle in a nitrogen environment; the mixture was stirred by a magnetic stirrer at 60 °C. Subsequently, DBTDL catalyst (0.3 g) was added into the four-necked bottle for a simultaneous reaction to form a prepolymer. Following 2 h of reaction, HBNPSi was slowly dripped into the four-necked bottle. After 2 h of reaction, 1,4-BD (0.8 g) was also allowed to react for 2 h and we observed whether viscosity increased. After viscosity increased for any sample, that sample was poured into a polytetrafluoroethylene mold and it was placed into a vacuum oven for 24 h of deaeration. After the temperature reached 60 °C, it was placed in another circulating oven for 24 h. When the temperature reached 60 °C, the finished product was withdrawn and stored at room temperature for cooling down, thereby completing the preparation of PU/HBNPSi. Its reaction formula is presented in Scheme 2. 

### 2.4. Measurements

The FTIR spectra of the materials were recorded between 4000 and 400 cm^−1^ using a Nicolet Avatar 320 FTIR spectrometer, from the Golden Valley, MN, USA. UV-VIS spectra were tested on a Hitachi (Tokyo, Japan) U-3300 spectrophotometer, and the sample was prepared as a thin film on a glass substrate by spin coating. Thin films were prepared by the solution-casting method. The samples were treated at 180 °C for 2 h and then ground into fine powder. The thermal degradation of the composite was examined using a thermogravimetric analyzer (TGA) (Perkin Elmer TGA 7, Waltham, Massachusetts, U.S.) from room temperature to 800 °C at a rate of 10 °C/min under an atmosphere of nitrogen. The measurements were made on 6–10mg samples. Weight-loss/temperature curves were plotted. The LOI is defined as the minimum fraction of O_2_ in a mixture of O_2_ and N_2_ that will just support flaming combustion. The LOI test was performed according to the testing procedure of the ASTM D 2836 Oxygen Index Method, with a test specimen bar 7–15 cm long, 6.5 ± 0.5 mm wide, and 3.0 ± 0.5 mm thick. The sample bars were suspended vertically and ignited by a Bunsen burner. The flame was removed and the timer was started. The concentration of oxygen was increased if the flame on the specimen was extinguished before burning for 3 min or burning away 5 cm of the bar. The oxygen content was adjusted until the limiting concentration was determined. The vertical burning test was done inside a fume hood. Samples were held vertically with tongs at one end and burned from the free end. Samples were exposed to ignition source for 10 s, then they were allowed to burn above a cotton wool until both sample and cotton wool extinguished. Observable parameters were recorded to assess fire retardancy. The UL-94 test classifies the materials as V-0, V-1, and V-2 according to the time period needed before self-extinction and the occurrence of flaming dripping after removing the ignition source. V-0 is the most ambitious and desired classification.

## 3. Results and Discussion 

### 3.1. Characterization of PU/HBNPSi Hybrid

The spectrum depicted in Figure 1 exhibits the characteristic absorption peak of DOPO-BQ; the –OH functional group was between 3600 and 3200 cm^−1^ [9,10], the P–Ph was at 1442 cm^−1^ [17], the P=O was at 1200 cm^−1^, and the P–O–Ph was at 759 cm^−1^ [18]. This interacted with the –NCO functional group at 2270 cm^−1^ for the characteristic absorption peak of IPTS [19] and a ring-opening reaction with the oxirane ring functional group at 910 cm^−1^ for the characteristic absorption peak of TGIC [20,21]. It was evident from HBNPSi that the –NCO functional group had already disappeared; the –NH functional group was between 3400 and 3300 cm^−1^ of –NH [22]; the C–N functional group was at 1471 cm^−1^ [17]; other characteristic absorption peaks of the original DOPO-BQ, IPTS, and TGIC appeared. This proved that the reaction had successfully formed the HBNPSi flame retardant. 

Figure 2 explains a reaction of the –NCO functional group at 2270 cm^−1^ of the prepolymer [19], with the –OH functional group between 3600 and 3200 cm^−1^ of HBNPSi [15,16], and the –OH functional group with an added chain extender. The figure indicated that reaction of the –NCO functional group of the prepolymer was completed and had formed PU/HBNPSi. The original functional groups that appeared in Figure 2 were –NH, C=O, C–N, P=O, P–O–Ph, and Si–O–Si, which verified that the preparation of PU/HBNPSi hybrid material had been successful. 

### 3.2. Spectral Transmittance Analysis 

The PU/HBNPSi hybrid material prepared through the sol–-gel method in this study was subjected to UV-VIS testing in the visible range (λ = 400 to 800 nm) to determine its spectral transmittance. 

Figure 3 indicates that the UV-VIS spectra of various samples of PU/HBNPSi with different concentrations of HBNPSi had nearly 100% penetration within the visible range. This suggested that PU/HBNPSi hybrid material had excellent compatibility between the organic phase and the inorganic phase. Additionally, this indicated that flame-retardant particles contained in the hybrid material were smaller than visible optical wavelength. The transparency of the material exhibited basic characteristics of nanomaterials, which have a wide range of applications. 

### 3.3. Thermal Property 

The heating rate of TGA was set to 20 °C/min and measurement was conducted under a nitrogen environment. When various concentrations of HBNPSi flame retardant were mixed into PU samples, changes were found in weight loss when the temperature was increased; the results are shown in Figure 3 and Figure 4 and in Table 1.

Changes in the degradation curve can be observed from Figure 4 and Table 1. The weakest C–NH chain of pristine PU caused degradation of hard segments of PU, which led to the formation of isocyanates, –OH functional groups, –NH functional groups, and carbon dioxide [23]. The temperature was 302 °C when weight loss was 10% (T_d10_). For PU samples with a high concentration of HBNPSi flame retardant, the T_d10_ of PU/HBNPSi 40% increased to 326 °C. The char yield at 800 °C increased with an increase in the amount of HBNPSi added. Specifically, the char yield (CY) in pristine PU increased from 0.7 to 8.1 wt.%. This was because when HBNPSi underwent thermal degradation, its phosphorus tended to form a gas phase that captured free radicals, produced a condensed phase, and catalyzed the formation of char [24]. Additionally, due to surface migration phenomena [25,26], silicon dioxide structures formed a compact layer protecting the internal parts of the substrate. Benzene rings supplied the production of char and thus char layers had antioxidants that resisted high-temperature combustion. This indicated that adding HBNPSi increased the thermal stability of the materials. 

Figure 5 presents a curve diagram of DTG for various changes in temperature. The temperature of maximum degradation rate of pristine PU was 343 °C. The temperature of maximum degradation rate of PU/HBNPSi 40% increased to 376 °C in samples of PU with flame retardant. However, the maximum degradation rate of pristine PU was −36 (wt.%/min), but the value decreased to −17 (wt.%/min) for samples with HBNPSi flame retardant. Flame retardant can effectively reduce the rate of thermal degradation in materials and produce slower material collapse. Thus, if flame retardants are used in buildings and those buildings catch fire, the flame retardants increase the chance that residents and firefighters can escape from the burning building. Experimental evidence proved that adding HBNPSi flame retardant can effectively improve thermal stability. 

### 3.4. Integral Procedural Decomposition Temperature (IPDT)

IPDT is a general method for evaluating the thermal stability of materials. A* denotes the ratio of the area of the degradation curve derived from the overall experiment to the area of the overall TGA diagram in Figure 6, K* denotes the coefficient of A*, T_i_ denotes the initial temperature of the experiment, and T_f_ denotes the final temperature of the experiment. Two factors influenced IPDT value: (1) Temperature of initial degradation and (2) char yield. T_f_–T_i_ is the range of test temperature and T_i_ is room temperature; they are fixed for the test procedure in most cases. If char yield and initial degradation temperature are higher from A*, K* equation, it will let S_2_ become higher and S_1_ lower. A* and K* will increase, and IPDT will be higher. It means the materials have better thermal stability. A high IPDT value indicates a high thermal stability [27,28,29]; the calculation method is as follows [30]: 

The symbols’ meanings are as follows: 

IPDT (°C) = A* × K* × (T_f_ − T_i_) + T_i_

A* = (S_1_ + S_2_)/(S_1_ + S_2_ + S_3_)

K* = (S_1_ + S_2_)/S_1_

T_i_ = the initial experimental temperature

T_f_ = the final experimental temperature

S_1_, S_2_, and S_3_ are the areas in Figure 6.

IPDT values were calculated by substituting the obtained values of T_i_, T_f_, S_1_, S_2_, and S_3_ into the equation. IPDT values can be observed from Figure 7 and Table 1. The IPDT value of pristine PU was 348 °C. A sample with 40% HBNPSi flame retardant had an IPDT value of 488 °C. The result verified that IPDT value increased with an increase in concentration of flame retardant and thermal stability increased up to 140%. This explained why HBNPSi can increase the thermal stability of PU hybrid materials. In this study, the key factors are excellent compatibility between organic and inorganic phases and compact structure, which contributed to the formation of char. Char layers contained silicon dioxide, which formed antioxidants that were resistant against high-temperature combustion. Thus, composite materials had superior thermal stability under high temperature. 

### 3.5. TGΔ

To investigate the interaction effect between organic and inorganic phases in thermal degradation and combustion processes, TGΔ was calculated from the difference between weight losses of the polymer matrix and composites [31,32,33]:TGΔ (wt.%) = M_exp_ − M_the_:Curve of weight difference,(1)
where M_exp_ denotes the experimental value of TGA degradation in hybrid materials. M_the_ was calculated by using the substrate data and TGA degradation curves with different additive ratios. The interaction between organic and inorganic phases is very important for the composite system. If strong interactions exist, the inorganic filler will be distributed in the polymer matrix. The phenomena will let the composites possess better properties including thermal, flame retardancy, and mechanical properties. Figure 8 showed that hybrid materials mostly had positive deviation and this indicated that an interaction effect existed between pristine PU and the HBNPSi additive. Additionally, it also revealed that compatibility between pristine PU and HBNPSi was excellent for improving the thermal stability and flame retardancy of composite materials. 

### 3.6. Flame Retardancy of the Materials 

LOI was used to judge the level of flame retardancy of polymer materials from differences in oxygen and nitrogen concentrations. Oxygen accounts for 21% of the atmosphere and when the LOI is equivalent to 21 or below 21 some materials burn freely, such materials are flammable. Materials with LOI values between 22 and 25 tend to be extinguished spontaneously during combustion. Materials with LOI values greater than or equal to 26 are considered difficult to ignite [33]; their adjusted flow rate settings of oxygen and nitrogen were measured (mL/sec) and the corresponding equation is as follows:(2)$$\mathrm{LOI}=\frac{O_{2}}{O_{2}+N_{2}}\times 100.$$

Figure 9 and Table 2 show that pristine PU has an LOI index of 19 and thus can be designated as a highly flammable polymer material. When the flame-retardant concentration increased to 40%, the LOI value of the PU/HBNPSi hybrid material increased to 26 and flame retardancy increased by seven levels. From Table 2, LOI values increased with the increase of the contents of phosphorus and silicon. This demonstrated HBNPSi can improve flame retardancy of the PU hybrid, which is attributed to the fact that during the thermal degradation of HBNPSi, the phosphorus free radicals in the materials will be formed in a gas phase to capture hydrogen and hydroxyl free radicals and catalyze the formation of char in condensed phases [24] and lead to surface migration phenomena of silicon [25,26]. It also formed compact silicon dioxide structures that protected the internal parts of the matrix. Benzene rings supplied the production of char, and thus char layers had antioxidants that resisted high-temperature combustion.

In terms of the UL-94 test, pristine PU could not pass this flame-retardant test, meaning that polyurethane is highly flammable. When HBNPSi concentration reaches 30 wt.%, the UL-94 test result still showed a fail level. However, at 40% concentration, the results of UL-94 showed significant changes and was improved to V-2 grade. The flame retardant can stop the fire in 30 s, but the composites still have dripping phenomena.

### 3.7. Char Analysis by Raman Analysis

Raman analysis was used to detect the char obtained by placing PU/HBNPSi samples in an oven at 800 °C for either 1 min or for 5 min. Additionally, it was used to observe changes in fluctuations of the D-band (disordered band) and the G-band (graphitic band) about carbon structure. The D-band was located at approximately 1350 cm^−1^ and was composed of carbon structures of irregular aliphatic compounds with sp^3^ structure [34]; the G-band was located at 1580 cm^−1^ and was composed of aromatic substances or the so-called carbon structures of six-membered rings; it had sp^2^ graphite structure [30]. During combustion, the D-band was transformed to the G-band and the appropriate ratio can be obtained by dividing areas (D/G); this ratio can be used as an excellent indicator of carbonization level. 

Figure 10 and Figure 11 indicate changes in D-band and G-band among samples of PU/HBNPSi 10% and PU/HBNPSi 40% and results of the analysis are shown in Table 3. The table showed ratios of 2.45 and 1.33 for PU/HBNPSi 10% following 1 and 5 min of combustion, respectively. When PU/HBNPSi samples contained 40% flame retardant, ratios of 0.66 and 0.40 were found following 1 and 5 min of combustion, respectively. A decrease in G/D ratio indicated that graphite accounted for the majority of char. A large amount of additives corresponded to a highly obvious carbonization effect. The melting point of graphite is higher than 3000 °C; graphite is sufficient to withstand the temperature of a normal fire scene (normally 800 °C). If an additive can effectively increase the char content of materials during combustion, the additive can improve the thermal stability and nonflammability of the materials. 

## 4. Conclusions

FTIR was used to conduct structural characterization, to verify the successfully preparation the PU/HBNPSi hybrid materials. The whole synthesis took place in solvent and was kept in a liquid state. It is very easy to process, especially in coating application. TGA and LOI results showed that thermal stability and flame retardancy of pristine PU improved after HBNPSi had been added. Raman char analysis of PU/HBNPSi hybrid materials determined the correlation between char formation and thermal properties. Raman spectra proved that the amount of char in the PU samples increased as the amount of HBNPSi in the samples increased. After HBNPSi had been added, a sample’s production of char during combustion increased markedly; this phenomenon became more obvious as concentration increased and thus produced the effects of increased thermal stability and flame retardancy.

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
