# Peer review of "Preparation and Characteristics of an Environmentally Friendly Hyperbranched Flame-Retardant Polyurethane Hybrid Containing Nitrogen, Phosphorus, and Silicon"

_polymers, 2019, doi:10.3390/polym11040720_

Round 1

Reviewer 1 Report

General comments:

As a reviewer, I suggest that the paper entitled “Preparation and characteristics of an environmentally friendly hyperbranched flame retardant polyurethane hybrid containing nitrogen, phosphorus, and silicon” should be considered for publication in the Polymers only after major revision.

Authors should carefully correct all technical errors such as missing figure caption, figure numbering, using of abbreviations without definition of their meaning (like C.Y in Table 1. or DOPO-BQ in the first sentence of the abstract), presenting data in table before the corresponding discussion.

Authors should justify in more details the use of sol-gel technology and the presented precursor as well as such high applied concentration (40 %) of flame retardant under which the best LOI and thermal stability were achieved. Since that authors talk about flame retardancy and non-flammability, I propose to authors to include UL 94 testing of their samples and to add these results to discussion regarding the flammability and charring fenomenon. I also propose to authors to reveal how this high percentages of flame retardant influence mechanical properties of polyurethane.

Authors should enrich the scientific way of the results presentation and to build their discussion according to the obtained results avoiding excessive citation of the literature in the discussion. All those citations and the corresponding comments should find their place in the introduction. I also propose to authors to update the reference list with the reports from 2019.

Authors should explain more clearly the procedure for the “Integral procedural decomposition temperature” as well as what is the additional information obtained from this method.

What is the purpose of TGΔ? Which synergistic effect was analysed, since that only one flame retardant additive was used?

Authors should also correct the conclusion avoiding repeating the data from the abstract.

Author Response

Dear Reviewers:

Firstly, we would like to thank you for your comments and we have corrected the necessary text. The changes in the text have been highlighted in red.

Reviewer #1:

Q1: Authors should carefully correct all technical errors such as missing figure caption, figure numbering, using of abbreviations without definition of their meaning (like C.Y in Table 1. or DOPO-BQ in the first sentence of the abstract), presenting data in table before the corresponding discussion.

A1: Thanks for your kind suggestions. We have carefully rechecked the manuscript and corrected the technical errors.

Q2: Authors should justify in more details the use of sol-gel technology and the presented precursor as well as such high applied concentration (40 %) of flame retardant under which the best LOI and thermal stability were achieved. Since that authors talk about flame retardancy and non-flammability, I propose to authors to include UL 94 testing of their samples and to add these results to discussion regarding the flammability and charring phenomenon. I also propose to authors to reveal how this high percentages of flame retardant influence mechanical properties of polyurethane.

A2: Thanks for your kind suggestions. We have added the description of the sol-gel technology into the introduction section. Actually, we are preparing the next manuscript about the mechanical properties and flame retardancy of the composites after aging test. It will include UL 94 and tensile property of polyurethane composites.

Q3: Authors should enrich the scientific way of the results presentation and to build their discussion according to the obtained results avoiding excessive citation of the literature in the discussion. All those citations and the corresponding comments should find their place in the introduction. I also propose to authors to update the reference list with the reports from 2019.

A3: Thanks for your kind suggestions. We have enriched the content in the results and discussion section. We just used some citations to support our description. We have updated the references list from 2019.

Q4: Authors should explain more clearly the procedure for the “Integral procedural decomposition temperature” as well as what is the additional information obtained from this method.

A4: Thanks for your kind suggestions. From the definition of IPDT, IPDT( ℃)= A*×K*×(Tf -Ti)Ti

A* =(S1S2)/(S1S2S3)

K* = (S1+S2)/S1= 1+ S2/ S1

   Tf -Ti is the range of test temperature, Ti is room temperature, and they are fixed for the test procedure in most cases. If char yield and initial degradation temperature are higher from A*, K* equation, it will let S2 become higher and S1 lower. A* and K* will increase, and IPDT will be higher. It means the materials have better thermal stability. We have added the description in the text.

Q5: What is the purpose of TGΔ? Which synergistic effect was analysed, since that only one flame retardant additive was used?

A5: Thanks for your kind suggestions. The interaction between organic and inorganic phases is very important for the composite system. If strong interactions exist, the inorganic filler will be distributed in the polymer matrix. The phenomena will let the composites possess better properties including thermal, flame retardancy and mechanical properties. We use TGΔ to study the interaction effect. TGΔ was calculated from the difference between weight losses of polymer matrix and composites: TGΔ(wt%)=Mexp–Mthe. We have added the description into the section.

        In this section, we use synergistic effect because the flame retardant contains silicon, phosphorus and nitrogen elements. We think it is not good words to describe the interaction. It will produce the confusion. We have corrected the description in the section.

Q6: Authors should also correct the conclusion avoiding repeating the data from the abstract.

A6: Thanks for your kind suggestions. We have corrected the conclusion avoiding repeating the data from the abstract.

Reviewer 2 Report

Substance having phosphorus-bearing as well as silicon- and/or nitrogen-bearing substructures are gaining in importance as flame-retardants for polymeric materials. The study of Chiang et al. is a contribution to this field of research.

The authors synthesized a hyper-branched compound containing phosphorus, nitrogen as well as silicon using a three-step reaction sequence. However, in subsection 2.2, where the synthesis of the hyper-branched flame-retardant is described, not all required experimental details are provided. This subsection should be supplemented by information about substance isolation including the yield achieved, drying and substance properties (solid? liquid?).

The novel substance was incorporated into a polyurethane (based on isophorone diisocyanate) and its influence on the thermal properties as well as the flammability of the polyurethane was investigated. The additive was applied with loadings of 10-40 wt%. The authors should provide the phosphorus content as well as the contents of silicon at the individual loadings.

Thermogravimetric investigations revealed that incorporation of the flame-retardant increases the thermal stability and the char yield moderately. It is an important finding that the authors showed that the flame-retardant does not induce a serious deterioration of the transmittance for visible light. The flammability of PU samples were examined using the limiting oxygen index (LOI). The LOI values showed an increase from 19 (pristine PU) to 26 (PU containing 40 wt% of the flame-retardant). The LOI increase above 25 indicates a pronounced decrease of the flammability. However, a sufficient high LOI value could only be achieved in case of relatively high loading (40 wt%). The authors could show that the novel flame-retardant acts in the condensed phase by increasing the char formation. Unfortunately, it was not investigated, whether the flame-retardant has activity in the gas phase too (by radical processes known from structural similar phosphorus compounds). In this context, it is not clear, what following sentence means:  “… because in the process of thermal degradation, phosphorus and its structure tended to form a gas phase, prevent the existence of a condensed phase, catalyze the formation of char, …”. This sentence seems to me confuse and should been rewritten.

It is a pity that the authors did not investigate the influence of the flame-retardant on the processability of the isocyanate/polyol system and the crucial mechanical parameters of the PU because additives impair these properties in many cases.

Nevertheless, the manuscript includes interesting findings and therefore I recommend it for publication in polymers. However, the alterations and supplementations mentioned above have to be carried out before publication. In addition, there is a few errors in spelling and grammar within the presented manuscript. Therefore, the authors should check the text again.

Author Response

Dear Reviewers:

Firstly, we would like to thank you for your comments and we have corrected the necessary text. The changes in the text have been highlighted in red.

Reviewer #2:

Q1: The authors synthesized a hyper-branched compound containing phosphorus, nitrogen as well as silicon using a three-step reaction sequence. However, in subsection 2.2, where the synthesis of the hyper-branched flame-retardant is described, not all required experimental details are provided. This subsection should be supplemented by information about substance isolation including the yield achieved, drying and substance properties (solid? liquid?).

A1: Thanks for your professional suggestions. We used THF as the solvent to synthesize HBNPSi flame retardant. After the reaction, we did not vaporize THF. The flame retardant was dissolved in the solvent, and then take place the chemical reaction with prepolymer to form the nanocomposites. Because the synthesis of flame retardant is addition reaction, the yield of the reaction is 100%. We have added some description into the subsection.

Q2: The novel substance was incorporated into a polyurethane (based on isophorone diisocyanate) and its influence on the thermal properties as well as the flammability of the polyurethane was investigated. The additive was applied with loadings of 10-40 wt%. The authors should provide the phosphorus content as well as the contents of silicon at the individual loadings.

A2: Thanks for your kind suggestions. We have provided the phosphorus content as well as the contents of silicon at the individual loadings in Table 1.

Q3: Thermogravimetric investigations revealed that incorporation of the flame-retardant increases the thermal stability and the char yield moderately. It is an important finding that the authors showed that the flame-retardant does not induce a serious deterioration of the transmittance for visible light. The flammability of PU samples were examined using the limiting oxygen index (LOI). The LOI values showed an increase from 19 (pristine PU) to 26 (PU containing 40 wt% of the flame-retardant). The LOI increase above 25 indicates a pronounced decrease of the flammability. However, a sufficient high LOI value could only be achieved in case of relatively high loading (40 wt%). The authors could show that the novel flame-retardant acts in the condensed phase by increasing the char formation. Unfortunately, it was not investigated, whether the flame-retardant has activity in the gas phase too (by radical processes known from structural similar phosphorus compounds). In this context, it is not clear, what following sentence means:  “… because in the process of thermal degradation, phosphorus and its structure tended to form a gas phase, prevent the existence of a condensed phase, catalyze the formation of char, …”. This sentence seems to me confuse and should been rewritten.

A3: Thanks for your kind suggestions. We did not investigate the mechanism of flame retardancy in gas phase. The behavior of free radicals could be studied through Py-GC/MS analysis. We will do this experiment in next manuscript to investigate the mechanism.

We have rewritten the sentence. “This demonstrated HBNPSi can improve falme retardancy of PU hybrid, which is attributed to the fact that during the thermal degradation of HBNPSi, the phosphorus free radicals in the materials will be formed in a gas phase to capture hydrogen and hydroxyl free radicals and catalyze the formation of char in condensed phases”. We have added it into the context.

Q4: It is a pity that the authors did not investigate the influence of the flame-retardant on the processability of the isocyanate/polyol system and the crucial mechanical parameters of the PU because additives impair these properties in many cases.

A4: Thanks for your kind suggestions. The whole synthesis took place in solvent and kept in liquid state. It is very easy to process, especially in coating application. We have added the description in conclusion section.

Actually, we are preparing the next manuscript about the mechanical properties of the composites after aging test. It will include tensile property of polyurethane composites.

Q5: Nevertheless, the manuscript includes interesting findings and therefore I recommend it for publication in polymers. However, the alterations and supplementations mentioned above have to be carried out before publication. In addition, there is a few errors in spelling and grammar within the presented manuscript. Therefore, the authors should check the text again.

A5: Thanks for your kind suggestions. We wholly agreed with your professional advices about the manuscript and provided many additional descriptions. We also have carefully rechecked the manuscript and corrected some errors in the text.

Round 2

Reviewer 1 Report

Authors only partly answered to my questions and suggestions. So, I suggest to authors to check my comments once again in order to prepare the manuscript that will meet the Polymers journal criteria.

Author Response

Dear Reviewers:

Firstly, we would like to thank you for your comments and we have corrected the necessary text. The changes in the text have been highlighted in red.

Reviewer #1:

(The responses to Second Round)

Q1: Authors only partly answered to my questions and suggestions. So, I suggest to authors to check my comments once again in order to prepare the manuscript that will meet the Polymers journal criteria.

A1: Thanks for your patience. We have reviewed your questions and suggestions one by one and done our best to answer them. We hope the manuscript can meet the Polymers journal criteria.

(The updated responses to First Round)

Q1: Authors should carefully correct all technical errors such as missing figure caption, figure numbering, using of abbreviations without definition of their meaning (like C.Y in Table 1. or DOPO-BQ in the first sentence of the abstract), presenting data in table before the corresponding discussion.

A1: Thanks for your kind suggestions. We have carefully rechecked the manuscript and corrected all the technical errors such as missing figure caption, figure numbering, using of abbreviations without definition of their meaning.

Q2: Authors should justify in more details the use of sol-gel technology and the presented precursor as well as such high applied concentration (40 %) of flame retardant under which the best LOI and thermal stability were achieved. Since that authors talk about flame retardancy and non-flammability, I propose to authors to include UL 94 testing of their samples and to add these results to discussion regarding the flammability and charring phenomenon. I also propose to authors to reveal how this high percentages of flame retardant influence mechanical properties of polyurethane.

A2: Thanks for your kind suggestions. We have added the description of the sol-gel technology into the introduction section. We have added the UL-94 test into discussion section. In terms of UL-94 test, pristine PU could not pass this flame retardant test, meaning that polyurethane is highly flammable. When HBNPSi concentration reaches 30 wt%, the UL-94 test result still showed fail level. However, at 40% concentration, the results of UL-94 showed significant changes and was improved to V-2 grade. The flame retardant can stop the fire in 30 seconds, but the composites still have dripping phenomena.

        Polyurethane is one kind of rubber and mechanical properties were obviously affected by adding the fillers. For example, the tensile strength and modulus will increase with the increasing of the additives, but the elongation at break will decrease through the addition of the filler. Actually, we are preparing the next manuscript about the mechanical properties of the composites after aging test. It will include tensile property of polyurethane composites.

Q3: Authors should enrich the scientific way of the results presentation and to build their discussion according to the obtained results avoiding excessive citation of the literature in the discussion. All those citations and the corresponding comments should find their place in the introduction. I also propose to authors to update the reference list with the reports from 2019.

A3: Thanks for your kind suggestions. We have enriched the content in the results and discussion section. We just used some citations to support our description in discussion section. We have updated the references list from 2019.

Q4: Authors should explain more clearly the procedure for the “Integral procedural decomposition temperature” as well as what is the additional information obtained from this method.

A4: Thanks for your kind suggestions. From the definition of IPDT, IPDT( ℃)= A*×K*×(Tf -Ti)Ti

A* =(S1S2)/(S1S2S3)

K* = (S1+S2)/S1= 1+ S2/ S1

   Tf -Ti is the range of test temperature, Ti is room temperature, and they are fixed for the test procedure in most cases. If char yield and initial degradation temperature are higher from A*, K* equation, it will let S2 become higher and S1 lower. A* and K* will increase, and IPDT will be higher. It means the materials have better thermal stability. We have added the description in the text.

Q5: What is the purpose of TGΔ? Which synergistic effect was analysed, since that only one flame retardant additive was used?

A5: Thanks for your kind suggestions. The purpose of TGΔ is to show the interaction between polymer matrix and the fillers. The interaction between organic and inorganic phases is very important for the composite system. If strong interactions exist, the inorganic filler will be distributed in the polymer matrix. The phenomena will let the composites possess better properties including thermal, flame retardancy and mechanical properties. We use TGΔ to study the interaction effect. TGΔ was calculated from the difference between weight losses of polymer matrix and composites: TGΔ(wt%)=Mexp–Mthe. We have added the description into the section.

        In this section, we use synergistic effect because the flame retardant contains silicon, phosphorus and nitrogen elements. We think it is not good words to describe the interaction. It will produce the confusion. We have corrected the description in the section.

Q6: Authors should also correct the conclusion avoiding repeating the data from the abstract.

A6: Thanks for your kind suggestions. We have corrected the conclusion avoiding repeating the data from the abstract.

Round 3

Reviewer 1 Report

Authors reviewed my comments and included the suggested UL 94 measurements in the manuscript. Since that authors said that they will include the tensile properties of the polyurethane composites in the next manuscript, I suggest that the paper entitled “Preparation and characteristics of an environmentally friendly hyperbranched flame retardant polyurethane hybrid containing nitrogen, phosphorus, and silicon” can be considered for publication in the Polymers.